# Insights into the Molecular Basis of Pollen Coat Development and Its Role in Male Sterility

**DOI:** 10.3390/ijms26157036

**Published:** 2025-07-22

**Authors:** Binyang Lyu, Cuiyue Liang

**Affiliations:** 1College of Life Science, South China Agricultural University, Guangzhou 510642, China; m15606946826@163.com; 2College of Resources and Environment, South China Agricultural University, Guangzhou 510642, China

**Keywords:** biosynthesis, fatty acids, function, humidity-sensitive genic male sterility

## Abstract

The pollen coat is the outermost layer of pollen and plays a key role in successful pollination and environmental adaptation. It consists of lipids, proteins, and phenolic compounds that protect pollen from environmental stress, promote hydration, and enable a proper interaction with the stigma. However, many questions remain unanswered, such as what the components of the pollen coat are and how they are formed, as well as how defects in the pollen coat affect the normal function of pollen. This review highlights the molecular mechanisms behind the biosynthesis and transport of pollen coat components and their contributions to pollen hydration, pollination compatibility, and fertility. Moreover, we discuss the role of selected gene families in pollen coat formation and their potential impact on agricultural breeding, paving the way for the breeding of more efficient crops.

## 1. Introduction

The pollen coat is composed of lipids, proteins, and phenols and is essential for angiosperm pollen [1,2]. It participates in plant reproduction and adaptation to the environment. First, the pollen coat protects the pollen from environmental stresses, such as ultraviolet radiation, drought, and temperature changes, during the period from anther dehiscence to pollination, allowing the pollen to maintain viability for germination and complete the pollination process [3,4,5]. Research on the proteomes of pollen coats from different plants emphasizes the important role of protein components in these functions [6]. Second, the pollen coat is involved in pollen–stigma interactions. During the early pollination process, the stigma has the ability to identify the pollen molecule, thus determining the compatibility of the pollen; therefore, only suitable pollen can fertilize. This recognition system prevents self-pollination and promotes cross-pollination [7,8,9]. The interaction mechanism includes specific molecules that are present in the pollen coat. This approach helps identify the signal transmission process between pollen and stigma, which serves as the foundation for developing improved pollination techniques. In the study of the pollen coat components of Brassicaceae, two main factors are known to play a role in pollen and stigma interactions: the physical characteristics of the pollen coat and the signal transduction [10,11,12]. Additionally, several mutants lack signaling or receptor factors, preventing proper molecular recognition and causing the pollen to be rejected. Research on *Arabidopsis thaliana* shows that defective pollen coats tend to cause humidity-sensitive genic male sterility (HGMS), which is when plants are sterile in low-humidity environments and fertile in high-humidity environments [10,12,13].

The application value in agriculture of research on the pollen coat is increasing. Firstly, the analysis of its composition, along with the communication between the pollen and stigma, allows us to choose crop parents with increased pollination success, which leads to higher yields [14,15]. Secondly, studying the external components of the pollen coat can help us further understand how to maintain pollen activity in adverse conditions, ensuring smooth pollination and providing theoretical support for the development of varieties with stronger tolerance to adverse environments [16,17]. Finally, the study of the pollen coat can help minimize labor costs for pollination. While artificial pollination can be effective for crop yield enhancement, this method is not suitable for large-scale crops, because it is very time-consuming and expensive [18]. Through the study of the coating of pollen, we can improve the seed setting rate of crops under natural conditions; this type of research can also provide a large number of conditionally sterile materials for hybridization, thus reducing the cost of crop production.

Currently, there are a number of applications of pollen coat research in botany and agriculture, but many unknowns remain. For example, the transportation mechanism of pollen coat precursors after synthesis has not been fully studied, and the specific proportion that is required for them to maintain pollination remains unclear. In addition, researchers have not yet fully understood the molecular process that explains how pollen coat precursors are synthesized and transported [3]. It is therefore necessary to investigate these questions further in order to better understand the transportation process of the pollen coat and the molecular mechanisms of each component during the pollination process. In this review, we summarize the current discoveries in the field of pollen coat research, including its composition, functions, and known transport pathways of certain precursors, providing research ideas for future studies.

## 2. Pollen Coat Formation and Morphology of Pollen in Arabidopsis and Rice

### 2.1. Pollen Coat Development and Formation

The formation and development of the pollen coat are key steps in the maturation process of pollen grains, directly related to tapetal cell secretory activity and surface substance deposition [19,20]. This process unfolds across defined developmental stages: Initially, during the tetrad stage (stage 8), the tapetal layer secretes callase, which degrades callose, enabling the release of microspores from tetrads and initiating the deposition of the primexine [3,21]. Subsequently, in the free microspore stage (stage 9), the exine of pollen grains continues to develop, and the intine begins to form. Next, during the vacuolate micropore stage (stage 10), PCD signaling is activated, and the tapetum layer begins to degrade. As development progresses to the bicellular pollen grain stage (stage 11), the pollen coat gradually forms [3,20,22]. Finally, in the tricellular pollen grain stage (stage 12), pollen grains are fully mature with the pollen coat accumulation complete, making them ready for pollination (Stages 1–7 are primarily focused on the formation of the anther locule and microsporogenesis. As these stages are not directly related to the formation of the pollen coat, they are not discussed in detail.) [3,20,23]. When pollen matures, complete tapetal degradation yields a highly hydrophobic pollen coat, which confers environmental stress resistance, this process is regulated by multiple genes and hormones [3,24,25,26]. Transcription factors modulate tapetal development and secretion, thereby indirectly controlling pollen coat formation [20,27,28]. Meanwhile, hormones (such as brassinosteroids and jasmonic acid) regulate tapetal secretory activity and gene expression, ultimately defining the pollen coat composition and structure [19,29]. These mechanisms are essential, not only for pollen maturation and protection but also for successful pollination and fertilization.

### 2.2. Pollen Morphology

Although pollen’s morphology is significantly different among plant species, the pollen wall is generally composed of three structural components: the intine, exine, and pollen coat [19,30] (Figure 1). The innermost intine has a bilayer structure consisting of the exintine and endintine. Hydrophobic proteins, pectin, hydrolytic enzymes, cellulose, and hemicellulose are the main components of the intine and are essential for facilitating pollen germination, controlling the growth direction of pollen tubes, and retaining structural integrity [31,32,33,34].

The exine is composed of a sexine (the outer layer) and a nexine (the inner layer), with the endexine being divided into Nexine I and Nexine II [35,36]. The sexine is composed of two layers: the tectum and baculum [37,38,39]. It features chemical stability and mechanical robustness and mainly consists of sporopollenin, a resistant biopolymer that is composed of covalently crosslinked phenylpropanoid and lipid monomers through ether and ester linkages [40].

The development of microspores in rice reveals that sporopollenin gradually accumulates, covers the exine structure, and reinforces it. With the accumulation of sporopollenin and the subsequent deposition of the pollen coat, the pollen exine develops fully [39,41]. However, the morphology and composition of pollen’s outer wall are species-dependent [6,39,42]. *Arabidopsis* exine has a longer baculum, semi-open tectum, and thin nexine layer; the sculptured cavities of the exine are filled with an abundant pollen coat [4,19,37,43]. Rice has a higher bacula density, thicker nexine and tectum, and a much thinner pollen coat than *Arabidopsis* [3,43,44,45]. Moreover, compared to the carved exine of *Arabidopsis*, rice pollen has a smooth exine. This variation could be due to the difference in pollination approaches.

Some studies point out that the structure of pollen undergoes certain changes when pollen grains lack water to help the pollen change shape and reduce water loss (Harmomegathy). The pollen wall has thinner areas, which are supported by relatively thicker endexines or intine [46,47]. When pollen dehydrates, the difference between these thin and thick regions causes the pollen wall to shrink and expand, leading to the folding of pollen grains. This folding allows the pollen to survive for a period of time after dehydration and quickly regain activity after rehydration to complete pollination.

## 3. The Main Components of Plant Pollen Coats

The main components of the pollen coat—lipids and their derivatives—are de novo synthesized from the tapetum. This biosynthesis pathway is partly similar to that of sporopollen. In this paper, we discuss these component precursors.

### 3.1. Phenolic Biosynthesis

The biosynthesis of flavonoids and other phenolic components occurs in the phenylpropanoid pathway, a secondary metabolic pathway that differs from the shikimic acid pathway. Phenolic compounds, including flavonoids (e.g., flavonol, flavone, flavanone, flavonoid, isoflavones), lignins, stilbenes, and coumarins, which are produced by the phenylpropanoid pathway, serve as important components of the exine and tryphine [48,49,50]. In the ER of *Arabidopsis*, fatty acids are modified in a series of reactions involving hydroxylation, condensation, and reduction [51,52,53]. Proteins involved in these processes promote the production of lipids and phenolic precursors within the endoplasmic reticulum [54].

Studies have shown that derivatives of ferulic and *p*-coumaric acids are specific to the pollen coat. Coumaric acid 3-hydroxylase (C3H) catalyzes the conversion of *p*-coumaric acid to caffeic acid, which is then converted to ferulic acid by caffeic acid O-methyltransferase (COMT) [55]. Under the catalysis of 4-coumaroyl:CoA ligase (4CL), ferulic acid is converted into feruloyl-CoA [56], which is subsequently converted into N^1^, N^5^, N^10^-triferuloyl spermidine through the action of spermidine hydroxycinnamoyl transferase (SHT) [57,58]. Next, under the catalytic action of CYP98A8/9 and AtTSM1, hydroxycinnamoyl spermidine (HCAA) is synthesized [3,49,58]. *p*-Coumaroyl-CoA can also be converted to flavonols through a series of enzymatic reactions. First, *p*-coumaric acid is converted to *p*-coumaroyl-CoA by 4-coumaroyl-CoA ligase (4CL). Then, *p*-coumaroyl-CoA condenses with three molecules of malonyl-CoA, catalyzed by chalcone synthase (CHS), to produce chalcone. Chalcone is then converted to flavanone by chalcone isomerase (CHI). Flavanone is hydroxylated to dihydroflavonol by flavanone 3-hydroxylase (F3H). After that, dihydroflavonol is converted to flavonol via flavonol synthase (FLS) [59,60]. Finally, flavonols form flavonol 3-O-glucosyl glucoside through the action of glycosyltransferase UGT79B6/31 [61,62] (Figure 2).

### 3.2. Sterol and Triterpenoid Biosynthesis

Sterols and triterpenoids in the pollen coat facilitate the adaptation of plants to the environment. Both compounds are derived from the mevalonate (MVA) pathway [63,64], which is a common pathway that is shared by many different eukaryotes [65]. The biosynthesis of sterols commences with the conversion of acetyl-CoA, which is first converted to acetoacetyl-CoA by acetyl-CoA acetyltransferase (ACAT). Subsequently, it is converted to HMG-CoA by HMG-CoA synthase (HMGCS) and is finally converted to mevalonic acid by the action of HMG-CoA reductase (HMGCR) [66]. This intermediate is slowly transformed into two important molecules, isopentenyl pyrophosphate (IPP) and dimethylallyl diphosphate (DMAPP) [67,68,69]. The cyclization of IPP and DMAPP eventually leads to the biosynthesis of squalene [70,71], an important step in sterol biosynthesis that is catalyzed by the enzyme squalene synthase (SQS). By means of the action of squalene epoxidase (SQE) and Cycloartenol synthase (CAS), squalene is converted into 2,3-Oxidosqualene and then to cycloartenol [72,73], which is an important intermediate product in the synthesis of sterols and other isoprenoid molecules in plants [74,75]. This conversion process includes cyclization, hydroxylation, and rearrangement. After the formation of cycloartenol, it is further modified to produce different phytosterols by means of the action of enzymes [45,76] (Figure 2).

*FLAKY POLLEN1 (FKP1)* in *Arabidopsis* encodes an HMGS that is expressed in both the tapetum and microspores. The *fkp1* mutation causes abnormal tapetosome and elaioplast development in the tapetum, as well as reduced oleosin and sterol esters in the pollen coat. *HMG1* is expressed in both the tapetum and microspores, and its mutant exhibits a marked reduction in the number of both elaioplasts and tapetosomes at stage 11, as well as pollen coat defects at stage 12 [77,78]. The double mutant *atipi1 atipi2* has a similar phenotype to *hmg1* [79]. However, the effects downstream of triterpene and sterol metabolism have only been studied in rice. *OsOSC12/OsPTS1* encodes 2,3-oxidized squalene cyclase, and its mutant exhibits a delay of liposome degradation in the tapetum, resulting in male sterility at a relative humidity below 60%. The stearic acid, palmitic acid, and linolenic acid levels in the pollen coat drastically decrease, causing pollen grains to dehydrate rapidly. However, the exogenous application of a mixture of linolenic, stearic, and palmitic acids can effectively inhibit pollen dehydration in the *ososc12* mutant [80].

### 3.3. Fatty Acid Biosynthesis

One of the lipid derivatives in the pollen coat is fatty acids, which are synthesized in the tapetum through de novo processes. VLCFAs have a carbon chain length exceeding 20 carbons and play an important role in the differentiation, division, and growth of plant cells [4,39,81,82]. VLCFA defects cause male sterility, the absence of wax, slow embryonic development, and possible death [4,17,39,81,82,83]. In plants, VLCFAs are less well studied than classic fatty acids. In many plants, fatty acid elongation stops at C18. However, in plants with well-developed wax and cuticle layers, VLCFAs can reach up to C38 [81,84]. Long-chain fatty acids are biosynthesized in a process that can be divided into two steps. The first step is the catalyzation of Acetyl-CoA with several enzymes, including fatty acid synthase (FAS), fatty acid acyl-ACP thioesterases A and B (FATA/B), and long-chain acyl-CoA synthases (LACS), to generate free C16 and C18 fatty acids. LACS1 is important in the synthesis of C16 and C30 fatty acids. LACS1 and LACS2 are functionally redundant in wax synthesis [85,86,87]. The second step occurs after fatty acyl-CoA is transported into the ER, at which point fatty acids with C18 are elongated into VLCFAs (C20–C34) by means of the action of enzymes. VLCFAs are components of the pollen coat, sporopollenin, and wax [33,81] (Figure 2).

CER1 is an aldehyde decarbonylase, which converts C28-C34 aldehydes to C27–C33 alkanes. CER1 is related to epicuticular wax development in the stem, as well as pollen fertility. Null mutations inhibit the formation of alcohols and ketones, thereby affecting the pollen–stigma interaction [88,89]. CER2 and CER2-LIKE are homologous to BAHD acyltransferases and are necessary for fatty acid elongation in *Arabidopsis* and rice [41,90]. These genes are involved in the synthesis of C28–C34 fatty acyl-CoA, a precursor of epidermal wax synthesis [91,92]. The *cer2* are defective in wax biosynthesis, particularly those with lipid chains that are longer than C28. In addition, *CER2* heterologous expression in yeast can change the acyl chain length between C28 and C30 [93]. CER2-like1 and CER2-like2 can affect the substrate specificity of condensing enzymes, while CER2-like3 and CER2-like4 are homologs of BAHD acyltransferase [93,94]. These proteins affect the acyl chain length and drive the production of wax and pollen coats [95,96]. Two CER2-like family members, CER26 and CER26-like, have been characterized in *Arabidopsis*; the null mutant exhibits defects in the wax synthesis of C ≥ 30, whereas complementation lines have rescued this phenomenon [92,97].

CER3 is a fatty acid hydroxylase that is expressed in the tapetum and microspores at anther stage 9. It plays a role in cuticle and wax biosynthesis and can reduce long-chain acyl CoA to the corresponding aldehyde [98]. In the pollen coat, the *cer3* mutant displays a larger number of smaller lipid droplets compared with the wild-type, further indicating a change in the composition of the coat. In addition, the null mutant is conditionally male sterile as a result of impaired pollen and stigma interactions. The smaller amount of wax on the mutant corroborates the critical role of long-chain fatty acids in successful pollination [99].

CER6/KCS6 is a member of the 3-ketoacyl-CoA synthase family, a key family for the synthesis and elongation of fatty acids. After the mutation of *cer6*, the wax on the plant epidermis disappears almost completely. Through an analysis of the mutant’s lipids, it was found that almost all fatty acids with carbon chain lengths exceeding 24 were significantly reduced compared with the wild-type, indicating that CER6 is essential for generating VLCFAs. The mutation of *cer6* can cause defects in the pollen coat, leading to male sterility. CER6 itself is mainly involved in the synthesis of C28 VLCFAs, but there are also reports that it can extend the length of the carbon chain to C30 in combination with CER2 [92,100]. CER6 has also been reported in tomatoes, and when compared to the wild-type, *sicer6* exhibits floral fusion at anthesis. Mutants exhibit male sterility, further emphasizing the importance of CER6 fertility [101,102]. In addition, KCS5 and CER6 have also been reported to have functional redundancy, as they are both responsible for the synthesis and extension of VLCFAs of C26–C30 [97].

The *WSL4* gene was isolated from rice and was shown to encode KCS in this plant. WSL4 is a homolog of CER6 in *Arabidopsis*. It is important for the elongation of VLCFAs beyond C22. *WSL4* overexpression can increase the wax content; however, the null mutant *wsl* exhibited decreased amounts of wax components that were longer than C28 and C30 [100]. The sterility of *osgl1-4* is probably related to the severe decrease in the amounts of C25 and C27 alkanes [103]. In addition, *hms1* is impaired in the biosynthesis of the C26 and C28 VLCFAs, which influence the formation of bacula and tryphine in the pollen wall, thus affecting the pollen’s ability to retain water and subsequently causing male sterility [16,17].

## 4. Transport Mechanism and Related Genes of Pollen Coat Precursors

Studies have shown that during pollen development, the synthesis and transport of lipids are essential to the development of the pollen coat and cuticles. The proper development of these two components directly affects the pollination progress. It is currently speculated that the development of the pollen coat may be regulated by the degradation of the tapetum. The SPL/NZZ-DYT1-TDF1-AMS-MYB103-MS1 pathway is conserved and has been shown to be related to both the degradation of the tapetum and the development of the pollen coat [104,105]. Any abnormality in the metabolism of this pathway can cause developmental defects, resulting in nuclear male sterility and microspore sterility. The transcription factors in this pathway are coordinately expressed during anther development, regulating the anther maturation, tapetum degradation, and pollen formation [106].

In *Arabidopsis*, *DYT1* and *AMS*, as helix–loop–helix (bHLH) transcription factors, are involved in the regulation of multiple genes related to the development of the pollen coat, tapetum, and pollen outer wall (such as *GRPs*, *UGT79B6*, *CYP98As*, *AtABCG26*, and *EXLs*) [107,108,109]. *MS1* can regulate the degradation of the tapetum layer and the expression of other genes that are involved in pollen coat formation. It is speculated that *MS1* is located upstream of other genes that are involved in pollen coat development, indirectly regulating its formation. *MS1* is expressed at the tetrad stage, while the formation of the pollen coat occurs at the bicellular pollen grain stage [110,111,112]. Studies also indicate that *MS1* can regulate At1g61110, MYB99, and At5g61430 to indirectly regulate lipid transporters, acyl lipids, phenylpropanoid metabolic pathways, and oleosin. Although both *TDF1* and *AtMYB103* encode *MYB* transcription factors, it is still unclear which specific genes they regulate in relation to the pollen coat [113].

Within tapetum cells, lipid transport is mediated by transporters on the plasma membrane. During the formation of tapetum plastids and pollen primexine, No Exine Formation 1 (NEF1), a plastid membrane transporter, regulates lipid accumulation and pollen sporopollenin deposition. The level of sterol esters is significantly reduced in the *nef1* mutant [36]. ISTL1 and LIP5 have functions in the secretion process of anther tapetum cells, while *istl1 lip5* hinders lipid transport and affects the formation of pollen walls. The abnormal transport and deposition of lipid components in mutant anthers lead to structural defects in the outer wall of pollen [114]. *atsec23ad* is a double mutant of coat protein complex II (COPII) and causes the formation of spike-like structures [115] on the pollen coat. SEC31B, another member of the COPII family, is involved in pollen coat formation. In the *atsec31* null mutant, the pollen coat has a linear, vesicular, low-electron-lucent appearance [115,116]. Fatty Acid Export 1 (FAX1) mediates the transfer of fatty acids between the plastid and the ER. Mutations of this gene lead to abnormal pollen coat deposition and altered fatty acid metabolism, resulting in a reduced pollination ability and seed set rate [117].

Transporters located on the plasma membrane export the pollen coat precursors from the tapetum cells. In *Arabidopsis*, *ABCG9* and *ABCG31* can transport steryl glycoside, which contributes to pollen coat formation and is produced by means of the catalytic action of UGT80A2 and UGT80B1. The *abcg9 abcg31* double mutant exhibits abnormal pollen coat deposition, with irregular structures in the pollen coat, which results in male sterility; however, the living rate of the pollen can be restored by increasing the humidity [118]. ABCG15 and ABCG26 are ABC transporters, while *abcg15* is male sterile and unable to form mature pollen and exhibits a significant decrease in wax components. The abnormal degradation of *abcg26*’s tapetum layer and the excessive accumulation of microspore lipids also cause male sterility [44,118,119]. TT12 and TT19 play a role in the transport of phenolic compounds. In *tt12* and *tt19*, flavonoid accumulation is severely reduced, causing a reduction in the number of flavonoids in the pollen coat [120]. AtNPF2.8 is a flavonoid glycoside transporter protein in the plasma membrane that transports flavonoid 3-O-sophoroside from the tapetum cells to the pollen surface [121]. In tapetal cells, *OsC6* encodes a lipid transporter with a lipid-binding ability. *OsC6* is involved in the transportation of wax. OsLTP47 and OsC6 interact to form a heterodimer, co-regulating lipid transport and possibly transporting pollen wall components to the pollen exine in a cooperative relay manner to regulate the transportation of pollen coat precursors [122,123].

Researchers have established that the tapetum layer provides most of the materials that develop into pollen coats. Organelles including elaioplasts and tapetosomes in *Arabidopsis* and Ubisch bodies, lipid bodies, and plastids in rice contain abundant pollen coat precursors [3,124,125]. This indicates that these organelles play a role in releasing metabolites during the formation of the pollen coat. The improper degradation of the tapetum can affect the formation of the pollen coat. During anther development in *Arabidopsis* (S8-S10), plastoglobules accumulate in elaioplasts within the tapetum cells. These elaioplasts contain abundant lipids, particularly sterol esters, glycolipids, phospholipids, and monogalactosyldiacylglycerols (MGDs). The lipids in elaioplasts are released in the maturing pollen grains, contributing to the formation of the pollen coat [79,126,127,128,129]. In stages 11 to 12, only the sterol esters remain stable, while in the final developmental stage of the anther, with the fragmentation of elaioplasts, sterol esters start to be released into the anther locules [129]. Tapetosomes, which contain phosphatidic acids, oleosin-like proteins, triacylglycerols, phosphatidylcholine, and phosphatidic acids, are secreted by the ER during stage 9. At stage 10, tubular elements containing flavonoids and alkanes aggregate in the tapetosomes [20,112,120,130]. By stage 12, as the tapetal cells degrade, tapetosomes are released into the anther locules. Simultaneously, the reduced triacylglycerols and oleosins are transported along with flavonoids to the surface of the mature pollen [120,129].

The pollen coat gradually forms as a filling material in the later stages of pollen development, and the development of the tectum and baculum directly affects the pollen coat. In some plants, pollen is highly conserved in the form of the pollen coat and sporopollenin. Three *Arabidopsis* genes, *MS2*, *CYP704B1*, and *CYP703A2,* and their homologous genes in rice, *OsDPW*, *OsCYP704B2*, and *OsCYP703A3* [131,132,133,134], are involved in the biosynthesis of sporopollenin. The mutations of these genes can lead to defects in the tectum, changes in the sporopollenin composition, or defects in the pollen coat [131,132,133,134,135,136,137].

## 5. Function of Pollen Coat

The pollen coat is a metabolite that is located on the outermost layer of pollen and plays an important role in pollen protection, adhesion, and water retention, as well as pollen–stigma communication during the reproductive process. In addition, pollen coat defects can lead to a low seed setting ratio and HGMS in *Arabidopsis* and rice [17,24,80,92,103].

### 5.1. Pollen Adhesion to Pollinators and Pollen Protection

The pollen coat is mainly composed of lipids, proteins, sugars, and phenolic compounds [111,113]. These components serve vital functions for pollen. The pollen coat contains a large amount of lipids, which can provide strong adhesion for the pollen coat and be useful for long-distance transmission. Simultaneously, it protects pollen against dehydration in dry conditions, thus extending its lifespan. Proteins can provide stability to pollen structures and protect them from UV and mechanical damage [138,139]. They can also interact with lipids and sugars to enhance pollen adhesion. Sugar mainly plays a role in hydration and adhesion in pollen. For transmission, sugars can help pollen stick onto the surface of pollinators such as insects, birds, and bats [140,141]. This stickiness is crucial for cross-pollination [142]. When the pollen absorbs water, the sugar in the pollen coat also improves the water retention ability and promotes pollen hydration [143,144,145]. Phenolic compounds, such as flavonoids and carotenoids, act as antioxidants. They can effectively resist the damage caused by oxidative environments, thus protecting pollen cells from the impact of free radicals and oxidative damage [146,147], and are crucial for maintaining pollen’s vitality during transportation, especially in harsh environments [148,149].

In addition, these components in the pollen coat provide chemical and physical barriers to resist microbial attacks, protecting pollen grains from bacteria, fungi, and other pathogens that may damage its integrity or interfere with fertilization [150,151,152].

### 5.2. Communication and Recognition Between Pollen and Stigma

The pollen coat is an important signal transmission medium between the stigma and pollen. Pollen coat recognition is indispensable in the pollination incompatibility response. This enables plants to distinguish between their own pollen and that of other plants, thereby promoting self-pollination or cross-pollination [153]. Recently, cysteine-rich peptides have been shown to be key components of pollen–stigma recognition, which is used to discriminate between a plant’s own pollen and that of other plants. Peptides such as the S-locus cysteine-rich protein (SCR) are essential to promote compatibility responses and interaction between the pollen and stigma [24,153]. In *Brassica*, SCR is part of the self-incompatibility system and plays a central role in pollen recognition [152,154]. When the pollen and stigma have the same S-haplotype, it can trigger self-incompatibility reactions and inhibit self-pollination, thus ensuring that plants avoid self-pollination. SCR is hydrophilic and can therefore be immobilized inside the hydrophobic pollen coat [155,156,157]. Once contact is made with the stigma, the pollen coat starts forming hydrophilic channels, enabling the transport of water from papilla cells to pollen grains. Then, SCR peptides and their homologous female receptor molecules, such as SLGs and S-site glycoproteins, are dissolved within aqueous spaces of the pollen coat [154]. This allows for the passage of multiple protein components between the pollen and stigma, facilitating their interaction. In addition to SCR, PCP-A1 can also interact with SLG according to reports on *Brassica oleracea* [24,102,158,159]. Both SLR1-BP1 and SLR1-BP2 are members of the *PCP-A* family, interacting with an S-locus-related glycoprotein 1 (SLR1) during the recognition process of pollen and stigma [102,159]. There are four members of the *PCP-B* family in *Arabidopsis*—PCP-Bα, β, γ, and δ—which mediate the adhesion of pollen and stigma. Among them, PCP-Bγ competes with the Rapid Alkalization Factor (RALF33) peptides in the stigma following pollination. PCP-Bγ reduces the reactive oxygen species content in the papilla cells, thereby facilitating mutual recognition with the stigma and allowing the pollen tubes to penetrate it [160,161].

The communication and recognition between pollen and the stigma provide significant potential for applications in agriculture and ecology: in plant breeding, an understanding of this recognition mechanism can improve hybrid breeding, breaking reproductive isolation and facilitating interspecific hybridization and thus providing a basis for the improvement of new varieties [162,163,164]. It also enables the use of related genes and molecular markers to accelerate the breeding process via assisted screening [165]. In agricultural manufacturing, understanding the characteristics of pollen–stigma communication would help us choose the most effective pollination time and method, thus improving the pollination efficiency and crop yield [14,166]. In terms of ecology and evolutionary biology, as a key part of reproductive isolation, pollen–stigma recognition aids in our understanding of plant evolution and species diversity formation [167]. Moreover, it provides a model for studying plant signal transduction and offers insights into the molecular mechanisms of plants’ responses to environmental and biological interactions.

### 5.3. Water Retention and Hydration Response

Unlike seeds, which have a long lifespan, pollen’s lifespan is measured in hours and requires insects for long-distance transmission. This requires a certain amount of water in the pollen to ensure that it survives and functions properly during pollination [32,168,169]. In fact, pollen contains less water than most plant tissues, and dehydration-related genes are also expressed in pollen. When pollen is excessively dehydrated, it can lead to pollen wall collapse [17]. The pollen coat directly covers the pollen grain surface, serving as an important protective barrier that helps the pollen retain moisture and avoid losing vitality during the critical period from anther dehiscence to successful pollination [170,171].

After pollination, free fatty acids in the pollen coat establish a water potential gradient between the pollen and stigma, assisting the growth of pollen tubes toward the ovules [3,113,172]. In rice, pollen dehydration is caused by a lack of free fatty acids such as linolenic acid, stearic acid, and palmitic acid and their triterpene esters. Alkanes such as C26 and C28 have been found to protect pollen from dehydration, while the alkanes C25 and C27 play a role in the adhesion process between the pollen and stigma [16]. In addition, alkanes and wax esters provide signal transduction during pollination to transport water from the stigma to the pollen [103].

In addition to its water retention, the pollen coat also promotes hydration when the pollen is in contact with the stigma. Previous studies have shown that a well-developed and mature pollen coat is essential for pollination. When genes affecting pollen coat development are mutated, it can lead to the pollen being unable to hydrate, resulting in male sterility [41,99,173]. However, this phenomenon can be reversed in high-humidity environments, indicating that the pollen coat plays a crucial role in the hydration process. The stigma of *Arabidopsis thaliana* is dry, meaning that under normal conditions, pollen must absorb water through the stigma to complete germination upon contact. During this process, the pollen coat undergoes reorganization. Within minutes of contact between the pollen and stigma, the pollen coat flows toward the contact surface, forming a “foot”-shaped structure that establishes strong adhesion between them [19,174]. Reorganization occurs in the “foot” area, which is essential for the pollination process. Through reorganization, hydrophobic pollen coats form water transport channels, enabling pollen hydration [156].

Currently, multiple genes have been found to be associated with the formation of the pollen coat and pollen hydration. CER1, CER3, and CER6 are related to lipid synthesis in the pollen coat, and their mutations can lead to a decrease in the VLCFAs in the pollen coat, thereby affecting the hydration and fertility of pollen [175]. Extracellular lipase 4 (EXL4), which regulates the permeability of the pollen coat and promotes pollen hydration [176], contains the predicted family II lipase domain, which is crucial for pollen hydration.

The wax layer on the outside of pollen and the inner layer are rich in long-chain fatty acids that have a dense structure; this can prevent bacteria and other pathogens from invading the inside of pollen, thus protecting its integrity. At the same time, flavonoids and other substances in the outer cover of pollen have antioxidant capacities, which can clear free radicals generated by ultraviolet radiation, prevent oxidative damage to pollen cells by free radicals, and maintain the stability of pollen.

### 5.4. Pollen Coat and Humidity-Sensitive Genic Male Sterility

In the later stage of pollen development, the tapetum releases the compounds of the pollen coat (such as lipids and proteins) through programmed cell death, filling the space within the baculum [19,129,177]. These compounds play a role in humidity adaptation. It should be noted that mutations in certain genes can affect the normal accumulation of these substances, causing developmental defects in the pollen coat and leading to HGMS (Figure 3).

In 1993, a mutant of CER6 was identified as HGMS, marking the first report of HGMS in *Arabidopsis thaliana* [10,178]. *cer6* is male sterile under low-humidity conditions but fertile when the humidity is >90% RH. The *cer6* mutant has a sparse pollen coat [10,13], but pollen from wild-type and supplemented lines both have a normal pollen coat that fills the carved pores within the baculum uniformly. In addition to *cer6*, other members of the CER family are also associated with HGMS. CER1 is an aldehyde decarbonylase, and its mutant has defects in the pollen coat, exhibiting male sterility under conditions of a relative humidity below 40% [16,178,179]. *cer3*, *cer2 cer2l2* double mutants, and *cer8 lacs4* double mutants are also associated with HGMS, and all of these mutants show defective pollen coats compared with the WTs [41,92,98,99,180]. In contrast, *faceless pollen-1* has a smooth pollen surface without an obvious carved exine. Studies have shown that smooth pollen is caused by the excessive accumulation of tryphine in the exine, which prevents pollen from germinating under low-humidity conditions [181,182].

In rice, genes that are linked to HGMS have also been reported and are associated with the synthesis and deposition of wax and VLCFAs in the pollen coat; these include *OsOSC12*, *OsGL1-4/OsCER1*, and *HMS1*. *ososc12/ospts1* is the first HGMS mutant to have been reported in rice, and it encodes a triterpene synthase [80]. This enzyme contributes to the biosynthesis of poaceatapetol, a compound that facilitates the esterification and deposition of VLCFAs on the pollen coat. *ososc12* mutants have reduced levels of long-chain fatty acids, sterols, and triterpene esters but elevated levels of certain plant sterols in their pollen coat. This results in poor adhesion and hydration of pollen under low-humidity conditions (RH < 60%) [80]. These results suggest that the ability of the pollen coat to adapt and respond to changes in humidity is an important aspect of regulating pollination.

*OsGL1-4/OsCER1* is expressed in pollen and tapetal cells. The *osgl1-4/oscer1* mutant displays another HGMS phenotype in rice, producing pollen that appears normal but rapidly dehydrates after flowering, followed by adhesion and hydration defects under low-humidity conditions. The mutant pollen is viable but exhibits sterility at normal humidity levels (30–60% RH), while its fertility is fully restored under high-humidity conditions (RH > 80%) [103,179] This mutation affects long-chain alkane metabolism, altering the lipid composition of the pollen coat and fertility.

*HMS1/OsCER2* encodes a β-ketoacyl-CoA synthetase. Along with the other HGMS genes in rice, the *hms1* mutant showed a high seed setting ratio under high-humidity conditions (90% RH) but was sterile under low-humidity conditions (45% RH), further proving that humidity is necessary for pollen fertility.

## 6. Diversity and Evolutionary Significance of Pollen Coat in Plants

Despite conserved primary components, distinct differences exist in the pollen coat across plant species (Table 1). This variation not only reflects differences in chemical composition but also mirrors the adaptive changes plants have undergone during evolution. In terms of chemical composition, the constituents of the pollen coat are different among species. For example, the pollen coat of Brassicaceae (consisting of *Arabidopsis thaliana* and *Brassica napus*) is rich in saturated sterol esters and phospholipids [37,152,156,183], whereas that of maize consists of more polysaccharides and proteins [184]. The pollen coat plays a role in pollen–stigma interactions across different plant species. However, its morphology and function also vary significantly among plants. For instance, in species such as those within the Brassicaceae family and *Oryza sativa*, the pollen coat forms a uniform lipid layer primarily responsible for maintaining pollen moisture [3,152]. In contrast, the Zingiberaceae has evolved sticky substances that assist in long-distance pollen transfer by pollinators [185].

In the process of plant evolution, the differences in the composition and characteristics of the pollen coat reflect the adaptive evolution of plants to various environmental conditions. Additionally, the pollen coat of sunflower is rich in fatty acids, which facilitate pollination in arid environments [186]. The composition of the cotton pollen coat varies substantially under extraordinary environmental conditions, contributing to high reproductive success [187]. The components of the rice pollen coat are critical for pollen hydration and protection, especially in low humidity conditions [3,16,17].

Additionally, the pollen coat is pivotal in determining whether a plant is self-compatible (SC) or self-incompatible (SI), specifically in sporophytic self-incompatibility (SSI) systems, which include Brassicaceae and Asteraceae [167,188,189]. Its critical role lies in delivering recognition cues: the pollen coat-specific S-locus cysteine-rich proteins SCR/SP11 engage in interactions with stigma papilla receptors, especially the S-receptor kinase SRK [152,190,191,192]. In SI plants, when the SCR/SP11 allele carried by the pollen coat matches the SRK allele expressed on the stigma of the same plant, a rapid signaling cascade is initiated [163,193,194]. This cascade culminates in callose deposition, an oxidative burst, and other defense responses that physically block pollen tube penetration into the stigmatic cuticle, thereby initiating self-rejection [163,194,195]. Thus, in SI species, the pollen coat serves as a direct signal source for initiating self-recognition and triggers rejection reactions [196]. By contrast, SC plants have dismantled this recognition-rejection mechanism. The key differences in the pollen coat composition lie in the protein profile and function: the pollen coat of SI species includes intact, functionally active SCR/SP11 proteins that effectively and specifically recognize the stigma SRK receptors [24], whereas those of SC plants lack functional SCR/SP11 due to gene deletions, loss-of-function mutations, or transcriptional silencing. Even if these proteins are present, defects in downstream stigmatic signaling components prevent incompatibility signal transduction, thereby permitting pollen tube germination and growth [163,192,194,197,198].

In summary, the variations in the pollen coat among different plants not only reflect the evolutionary characteristics of plants but also have profound implications for their reproductive strategies and adaptability. By studying the composition and function of the pollen coat, we can have a better understanding of the adaptive evolution of plants under different environmental conditions and the evolution of their reproductive strategies.

**Table 1 ijms-26-07036-t001:** Plants with studied pollen coat.

Species	Methodology	Author List
*Arabidopsis thaliana*	Molecular genetic analysis, proteomic analysis, cell biology observation, biochemical analysis	Shuaijie Wei, et al. [19], Ludi Wang, et al. [152], D. J. Murphy, [156]
*Arabidopsis lyrata*	Proteomic analysis	Ludi Wang, et al. [152]
*Caulokaempferia coenobialis*	Cytological analysis, RNA-seq, positive selection analysis	Guo Hui Lu, et al. [185]
*Hornstedtia hainanensis*		
*Pyrgophyllum yunnanense*		
*Zingiber nudicarpum*		
*Zea mays*	Multi-omics integration	Yanhua Li, Giampiero Cai, et al. [184]
*Oryza sativa* L.	Lipid analysis, molecular biology analysis, physiological analysis	Huiqiong Chen, et al. [17]
*Helianthus annuus* L. cv. Morden	Lipid and enzyme activity analysis	Shakya Rashmi, et al. [186], Sharma Basudha, et al. [199]
*B. oleracea*	Molecular genetic analysis, proteomic analysis, cell biology observation, biochemical analysis	Iwano Megumi et al. [183]
*Brassica napus*	Proteomic analysis, cell biology observation	Jiabao Huang, et al. [167]
*Brassica rapa*	Multi-omics integration, molecular genetic analysis	
*Brassica campestris*	Biochemical identification, proteomic analysis, molecular genetic analysis, cell biology observation	Takayama Seiji, et al. [159], Xueqing Liu, et al. [200]
*Gossypium hirsutum* L.	Cytological/morphological observation, transcriptome analysis, phytohormone analysis, bioinformatics analysis	Ji Liu, Chaoyou Pang, Shuxun Yu, et al. [201]
*Sorghum halepense*	Proteomic analysis, biochemical analysis, cell biology observation, molecular genetic analysis	Bashir, et al. [202]
*Phleum pratense*		
*Cynodon dactylon*		
*Secale cereale*	Proteomic analysis, cytological analysis	Kalinowski, et al. [203]
*Festuca pratensis*		
*Hordeum vulgare*	Proteomic analysis, molecular genetic analysis, biochemical analysis, cytological analysis	Suen, et al. [204]
*Raphanus sativus*	Molecular genetic analysis, gene expression analysis	Yung Geun Yoo, et al. [205]
*Picea pungens*	Biochemical identification, cell biology observation	Maria Breygina, Alexander Voronkov, Tatiana Ivanova, Ksenia Babushkina [206]
*Nicotiana tabacum*		
*Lilium longiflorum*		
*Solanum lycopersicum*	Cytological analysis, gene expression analysis, biochemical analysis	Anna Smirnova, Jana Leide, Markus Riederer [101]
*Lolium multiflorum*	Proteomic analysis, biochemical identification, cytological analysis	Andrzej Kalinowski, Marek Radlowski, Aleksandra Bocian [191]
*Citrullus lanatus*	Cytological observation, proteomic analysis, bioinformatic analysis	Chunhua Wei, et al. [207]

## 7. Conclusions and Future Prospects

### 7.1. Conclusions

The pollen coat is essential for angiosperm reproduction by mediating pollen–stigma recognition, hydration, and protection [208]. Its composition is dominated by lipids, including fatty acids (notably VLCFAs), chemical compounds (sterols and triterpenoids), and proteins. Pollen coat formation requires coordinated biosynthesis and transport regulated by the SPL/NZZ-DYT1-TDF1-AMS-MYB103-MS1 pathway, executed via lipid transporters, elaioplasts, and tapetosomes. Disruption of these processes causes pollination failure [19,107,209]. Studies of the HGMS genes in *Arabidopsis* and rice have revealed conserved molecular mechanisms, providing genetic tools to explore pollen coat function [10,13,16,17,88,93,96,210].

### 7.2. Future Prospects

However, several questions regarding the pollen coat need to be answered. For example, what are the key regulatory modules for pollen coat precursor synthesis? How is fertility restored in VLCFA-deficient pollen under high humidity? What are the precise transport mechanisms of precursors to the pollen surface? What proportion of pollen coat components must be maintained to ensure fertility? To answer these questions, cutting-edge techniques should be employed. First, multi-omics should be integrated to map regulatory networks and identify novel components. Secondly, dynamic profiling (mass spectrometry, fluorescence) could be used to link composition, function, and the stress response. Third, the subcellular transport pathways should be elucidated through compartment-specific probes and genetic/pharmacological validation (e.g., CRISPR). Fourth, coat composition could be optimized via metabolic engineering guided by multi-omics data analysis.

## Figures and Tables

**Figure 1 ijms-26-07036-f001:**
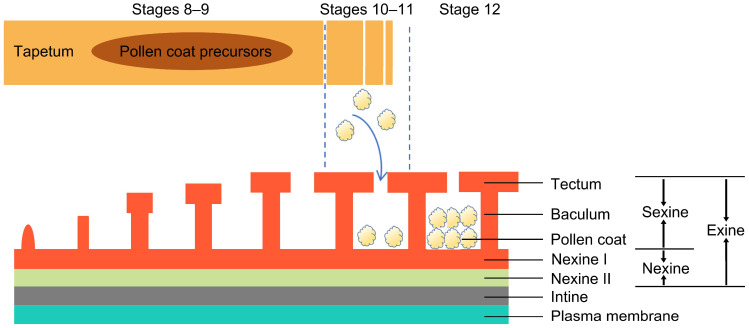
In *Arabidopsis*, the main material for the pollen coat comes from the tapetum layer. As the tapetal cells degrade, the pollen coat formation commences. By stage 12, pollen grains are fully mature with complete pollen coat accumulation.

**Figure 2 ijms-26-07036-f002:**
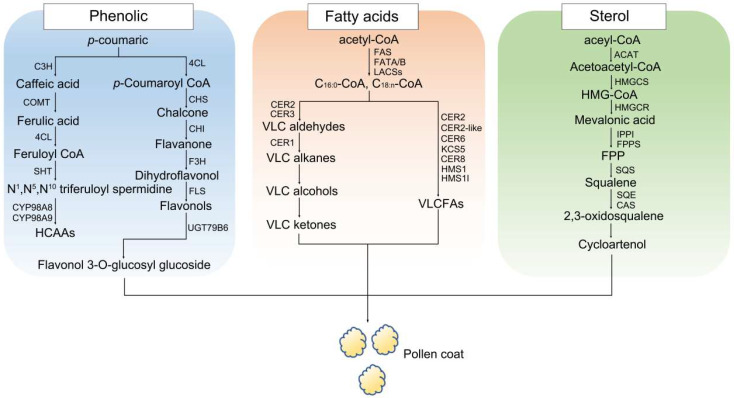
The biosynthetic process of the main components of the pollen coat. The synthesis of the main components of pollen coat precursors. Currently, the biosynthetic pathways of lipids, phenols, and sterols are relatively well studied, involving enzymes such as CAS (cycloartenol synthase), C3H (coumaric acid 3-hydroxylase), CHI (chalcone isomerase), CHS (chalcone synthase), 4CL (4-coumaroyl:CoA ligase), COMT (caffeic acid O-methyltransferase), F3H (flavanone 3-hydroxylase), FLS (flavonol synthase), HMGCS (HMG-CoA synthase), HMGCR (HMG-CoA reductase), IPP (isopentenyl pyrophosphate), IPPI (isopentenyl pyrophosphateΔ-isomerase), SQE (squalene epoxidase), SQS (squalene synthase), SHT (spermidine hydroxycinnamoyl transferase), FAS (fatty acid synthase), FATA (fatty acid acyl-ACP thioesterases A), FATB (fatty acid acyl-ACP thioesterases B), LACS (acyl-CoA synthase), and ACAT (acetyl-CoA acetyltransferase).

**Figure 3 ijms-26-07036-f003:**
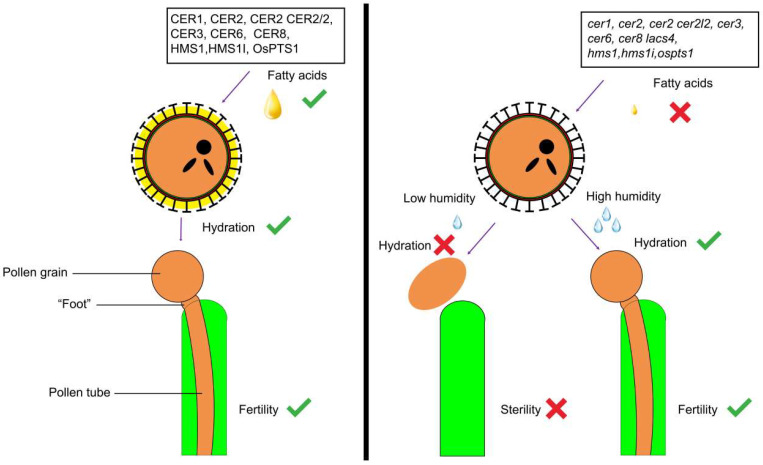
The development of the pollen coat affects pollen hydration. The genes that are currently known to cause HGMS are listed. In these mutants, the synthesis function of fatty acids and their derivatives is impaired, resulting in an incomplete pollen coat and sterility in low-humidity environments. However, under high-humidity conditions, even if the pollen coat is still incomplete, fertility can be restored.

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
