# Peer review of "Insights into the Molecular Basis of Pollen Coat Development and Its Role in Male Sterility"

_ijms, 2025, doi:10.3390/ijms26157036_

Round 1
Reviewer 1 Report
Comments and Suggestions for Authors
Dear authors,
I have read the manuscript under title ` Insights into the Molecular Basis of Pollen Coat Development and Its Role in Male Sterility`, written by Bin Yang Lyu and Cui Yue Liang.
I found the review very interesting and useful for breeders and anybody else who is interested in this topic.
It contains clear illustrations and the biosynthesis process of the main components of the pollen coat, which makes it easy to read and understand. It is well written and easy to read.
For a
better overview, the authors should create a table of all agricultural (and non-agricultural) plants where the pollen coat has been studied, with a list of authors. The table should have the methodology used for those studies.
The authors should underline the differences between the plant species. Is there a difference in pollen coat in plants/cultivars being self-compatible and self-incompatible. What is the implication in the evolution? This should be written in a separate section.
There are technical problems, some sentences are in
bold, some problems with small/big letters. References are not written according to the `Instruction for authors`. Latin names MUST be italic.
Author Response
Comments 1:「For a better overview, the authors should create a table of all agricultural (and non-agricultural) plants where the pollen coat has been studied, with a list of authors. The table should have the methodology used for those studies.]
Response 1: Thank you for pointing this out. We agree with this comment. As suggested, we have incorporated comparative studies on the pollen coat across additional plant species and compiled the data in a table. Concurrently, we introduced a dedicated section (Section 6) detailing interspecies variations in pollen coat characteristics and their evolutionary implications.- page 13-14.
Comments 2:「For a better overview, the authors should create a table of all agricultural (and non-agricultural) plants where the pollen coat has been studied, with a list of authors. The table should have the methodology used for those studies.]
Response 2: Thank you for pointing this out. We agree with this comment, we have verified the formatting of plant Latin names and ensured they are italic. Corrections were made to erroneous bold text (Page 8, Lines 2-4) and capitalization errors. Finally, the reference list was reformatted according to MDPI's most recent EndNote style template.
Note: All modifications have been highlighted in red.
Reviewer 2 Report
Comments and Suggestions for Authors
The manuscript “Insights into the Molecular Basis of Pollen Coat Development and Its Role in Male Sterility” is a good review paper. The pollen coat is the outermost layer of pollen and plays a key role in successful pollination and environmental adaptation. I think the manuscript can be better presented to the readers if the proper order is adjusted. As follow: 1. Introduction; 2. The development and formation of pollen and pollen coat; 3. Pollen coat morphology of different plants; 4. The main components of plants pollen coat; 5. The molecular mechanism of metabolism and synthesis of pollen coat; 6. Genes regulation of plant pollen coat; 7. Function of pollen coat: correlation between pollination, sterility and fertilityï¼›8. Conclusions and future prospects. The references should be united.
Author Response
Comments 1:「 I think the manuscript can be better presented to the readers if the proper order is adjusted. As follow: 1. Introduction; 2. The development and formation of pollen and pollen coat; 3. Pollen coat morphology of different plants; 4. The main components of plants pollen coat; 5. The molecular mechanism of metabolism and synthesis of pollen coat; 6. Genes regulation of plant pollen coat; 7. Function of pollen coat: correlation between pollination, sterility and fertility;8. Conclusions and future prospects.]
Response 1: Thank you for pointing this out. We agree with this comment. Based on your suggestions, we have reordered some sections of the manuscript, which has indeed improved its overall flow. In response to your recommendation regarding Section 2 ("The development and formation of pollen and pollen coat"), we have added new content to enrich this section (see the highlighted portion in Section 2.1 on page 2). As for your suggestion about Section 3 ("Pollen coat morphology of different plants"), we note that the pollen coat across different plant species is primarily composed of lipids covering the outermost layer of pollen grains, and thus shows no significant morphological differences. We thought you may referring to is the pollen exine. Therefore, we have merged your suggested Section 3 into Section 2, adding a dedicated subsection to describe it after introducing the formation process of the pollen coat.
Additionally, regarding your suggestion to separately introduce genes involved in regulating the pollen coat in Section 6, we attempted this approach but found that doing so would leave the previous two sections—composition of the pollen coat and transport mechanisms of pollen coat precursors—too thin to stand alone as full sections. Therefore, we integrated the gene-related information into the first two sections. This not only enriches their content but also helps readers better understand which genes play specific roles in different aspects of the pollen coat.
comments 2:「The references should be united]
Response 2: Thank you for pointing this out. We have thoroughly reviewed the manuscript and united the references.- page 10, paragraph 3. line 6.
Note: All modifications have been highlighted in red.
Reviewer 3 Report
Comments and Suggestions for Authors
I found this review as very interesting. It indicates the importance of pollen coat components on the pollen function, in particular authors highlathed its role in the first „pollen – stigma” reaction, i.e. in pollen hydration, pollen recognity (compatibility or incompatibility). They collected a lot of new biochemical and genetic information on the role of pollen coat. At the end, the authors raised many issues to be clarified, which is a positive side.
Major comment – the text lacks the Method section. The Authors should describe in details what method have been used to search in literaturÄ™ data, what detabases have been searched ?, what key words were applied?. How many papers were selected in the irst step, how many have beed rejected. What key was used to reject papers?
The procedure for searching should be described in details before paper publication.
Also, describe the particular steps (for example see Arksey and O’Malley Scoping studies: Towards a methodological framework. Int. J. Soc. Res. Methodol. 2005), e.g (i) article question formulation; (ii) recognition of relevant papers; (iii) selection and grouping of papers; (iv) data gathering, (v) manuscript drafting and final editing .
The Abstract is quity good, reflecting the most important conclusions from the literaturÄ™ searching. However, there is no information about the possible use of the collected information.
Introduction
It is quite good, covers most impotrant inforamtion, why authors undertaken study in the field of pollen coat research.
Introduction Lines 1-5 – te citiation is missing, add .
Methods – are missing. See above.
Text structure is clear, easy to follow, the information gathered are appropriate to the assumptions indicated in the title and introduction.
Fig 1 – The description have to be clarified. Please, add information what the stage 8-12 mean. Add the names. Add the information on the stages 1-7 – in short.
Section 6 Conclusions and future prospects. I suggest to separate future prospects and conclusion (It should give short opinion in 3-4 sentences).
Author Response
Comments 1:「Major comment – the text lacks the Method section. The Authors should describe in details what method have been used to search in literaturÄ™ data, what detabases have been searched ?, what key words were applied?. How many papers were selected in the irst step, how many have beed rejected. What key was used to reject papers?]
Response 1: Thank you for pointing this out. We agree with this comment. We regret this oversight, as this content was not observed in other published review articles within IJMS, we inferred it might require supplementary upload. Accordingly, we have attached the relevant file here.
comments 2:「Fig 1 – The description have to be clarified. Please, add information what the stage 8-12 mean. Add the names. Add the information on the stages 1-7 – in short.]
Response 2: Thank you for pointing this out. We have add the details for Stages 8-12 and provided a short information of Stages 1-7.-(page 3, figure legends part)
Comments 3:「Section 6 Conclusions and future prospects. I suggest to separate future prospects and conclusion (It should give short opinion in 3-4 sentences)]
Response 3: Thank you for pointing this out. We agree with this comment. We have separated the conclusion and future prospects sections, presenting both in a more concise manner. - page 14-15,paragraph 1-2.
Note: All modifications have been highlighted in red.
Round 2
Reviewer 1 Report
Comments and Suggestions for Authors
Dear authors,
I have read the corrected version.
It is improved but, I have a feeling that you have done these corrections a bit in a hurry, so it needs some fine tuning.
The table in the Section 6, has no number, no title. References within the table should be presented with numbers. You also talked in the text about tomato, but there is no tomato in this table. Besides, I have checked the literature, there are much more plants where this topic is already done, and they are not added. What about molecular base of self-incompatibility in many plant species? Please, this must be done.
The text under Figure 1 is too long. The title should be shorter, and the explanations should be within the text.
Author Response
Comments 1:「The table in the Section 6, has no number, no title. References within the table should be presented with numbers. ]
Response 1: Thank you for pointing this out. We agree with this comment. As suggested, we have added the table number, title and reference numbers.- page 13-15.
Comments 2:「Besides, I have checked the literature, there are much more plants where this topic is already done, and they are not added.]
Response 2: Thank you for pointing this out. This time, we thoroughly searched the database and added the additional plants into the table.- page 13-15.
Comments 3:「What about molecular base of self-incompatibility in many plant species?Please, this must be done.]
Response 3: Thank you for pointing this out. We have incorporated the established relationships and molecular mechanisms between pollen coat and self-compatible/incompatible plants.- page 13,paragraph 3.
Comments 4:「The text under Figure 1 is too long. The title should be shorter, and the explanations should be within the text.]
Response 4: Thank you for pointing this out. We have shortened the title and moved the corresponding explanations into the main text.- page 2,paragraph 4, the text highlighted in red.